# DialCoT Meets PPO: Decomposing and Exploring Reasoning Paths in Smaller Language Models

**Chengcheng Han**[◇*]    **Xiaowei Du**[♠]    **Che Zhang**[♡]
**Yixin Lian**[♠]    **Xiang Li**[◇]    **Ming Gao**[◇♣†]    **Baoyuan Wang**[♠†]

[◇]School of Data Science and Engineering, East China Normal University
[♠]Xiaobing.AI
[♡]School of Software & Microelectronics, Peking University
[♣]KLATASDS-MOE in School of Statistics, East China Normal University
chengchenghan@stu.ecnu.edu.cn
{duxiaowei,lianyixin,wangbaoyuan}@xiaobing.ai
mmt@stu.pku.edu.cn
{xiangli,mgao}@dase.ecnu.edu.cn

## Abstract

Chain-of-Thought (CoT) prompting has proven to be effective in enhancing the reasoning capabilities of Large Language Models (LLMs) with at least 100 billion parameters. However, it is ineffective or even detrimental when applied to reasoning tasks in Smaller Language Models (SLMs) with less than 10 billion parameters. To address this limitation, we introduce Dialogue-guided Chain-of-Thought (DialCoT) which employs a dialogue format to generate intermediate reasoning steps, guiding the model toward the final answer. Additionally, we optimize the model's reasoning path selection using the Proximal Policy Optimization (PPO) algorithm, further enhancing its reasoning capabilities. Our method offers several advantages compared to previous approaches. Firstly, we transform the process of solving complex reasoning questions by breaking them down into a series of simpler sub-questions, significantly reducing the task difficulty and making it more suitable for SLMs. Secondly, we optimize the model's reasoning path selection through the PPO algorithm. We conduct comprehensive experiments on four arithmetic reasoning datasets, demonstrating that our method achieves significant performance improvements compared to state-of-the-art competitors. [1]

## 1 Introduction

With the advent of Chain-of-Thought (CoT) prompting (Wei et al., 2022), which encourages Large Language Models (LLMs) to generate a series of intermediate steps to help get the final answer, reasoning capabilities of LLMs has seen

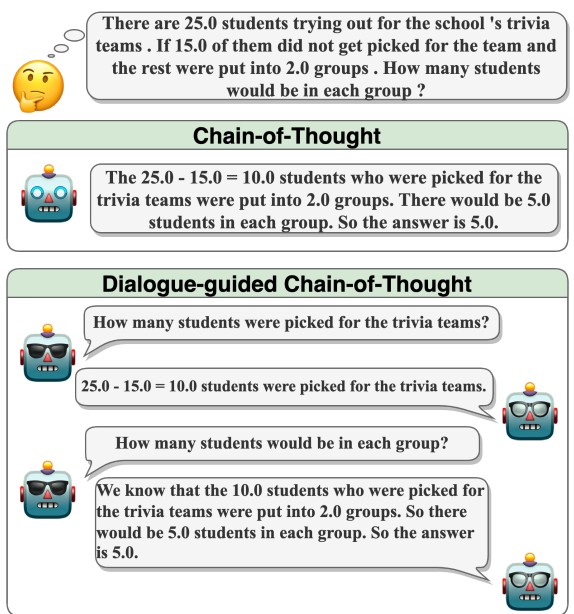

Figure 1: A comparison of DialCoT and CoT prompting (Wei et al., 2022). CoT prompting guides the model to output all intermediate reasoning steps at once to obtain the final answer, while DialCoT leads the model to gradually generate intermediate reasoning steps in a dialogic format to ultimately arrive at the answer.

significant improvement. However, preliminary results of Wei et al. (2022) have demonstrated that CoT prompting only shows significant performance gains on LLMs (≥100B), such as LaMDA-137B (Thoppilan et al., 2022), GPT-3 175B (Brown et al., 2020) and PaLM-540B (Chowdhery et al., 2022). But it is ineffective, or even detrimental, to the performance on reasoning tasks in Smaller Language Models (≤10B). This phenomenon is explained by Wei et al. (2022), who attribute it to abilities such as semantic comprehension and symbolic mapping, which only manifest at larger

---

[*]Work done during an internship at Xiaobing.AI.
[†]Corresponding author.
[1]Our code and dataset are publicly available at https://github.com/hccngu/DialCoT.

scales. The massive computational requirements and inference costs of LLMs make them unfeasible for widespread deployment. There is a pressing community interest in figuring out how to further enhance the reasoning capabilities within Smaller Language Models (SLMs).

Recent works (Magister et al., 2022; Ho et al., 2022; Fu et al., 2023) have attempted to enhance the performance of SLMs on reasoning tasks by fine-tuning them with training data generated from LLMs that contain intermediate reasoning steps. However, the results have been less than optimal. To further boost the reasoning capabilities of SLMs, we propose **Dial**ogue-guided **C**hain-**of**-**T**hought (**DialCoT**), which aims to progressively generate intermediate reasoning steps in the dialogue format, instead of generating all intermediate reasoning steps at once (as shown in Figure 1). Specifically, we assign to the model two roles: *Decomposer* and *Solver*. The *Decomposer* is tasked with breaking down the original question into a series of sub-questions. The *Solver* sequentially addresses each sub-question presented by the Decomposer, thereby obtaining the answer to the original question. They utilize different instructions while sharing the same model parameters. We propose three different forms of DialCoT: 1) **DialCoT-A** (**A**ll at once), in which the Decomposer generates all sub-questions at once and the Solver simultaneously provides all answers. 2) **DialCoT-M** (**M**ixed), where the Decomposer generates all sub-questions at once but the Solver sequentially delivers the answers of the sub-questions generated by the Decomposer. 3) **DialCoT-S** (**S**tep by step), where both the Decomposer and Solver operate sequentially to generate sub-questions and their corresponding answers. We provide a detailed comparison of the performance of the three different forms of DialCoT in Section 4.4. Furthermore, building upon DialCoT-S, we design **DialCoT-S-PPO**, which leverages the **P**roximal **P**olicy **O**ptimization algorithm to select the optimal reasoning path, thereby further enhancing its performance in reasoning tasks. Compared to previous methods (Ho et al., 2022; Fu et al., 2023), our approach has two main advantages:

1. We transform the process of solving a complex reasoning question into decomposing the question and solving a series of simpler sub-questions, which reduces the task difficulty and is more suitable for SLMs.

2. By breaking down intermediate reasoning steps into dialogue-formatted sub-questions and answers, we can use reinforcement learning more effectively to choose the optimal reasoning path from various options.

To validate the effectiveness of our approach, we fine-tune Flan-T5 (Chung et al., 2022) using 7000 training examples from the GSM8K (Cobbe et al., 2021) dataset that include intermediate questions and answers. The results surpass the latest method, SpecialFT (Fu et al., 2023), by 6.2%. Notably, the amount of training data we use is only 1/20 of that used by SpecialFT. In addition, to verify the model's generalization capability on out-of-distribution tasks, we also test on the MultiArith (Roy and Roth, 2015), ASDiv (Miao et al., 2020) and SVAMP (Patel et al., 2021) datasets. Our method achieves state-of-the-art performance compared to other baselines.

## 2 Related Work

### 2.1 Chain-of-Thought Prompting

Chain-of-Thought (CoT), which significantly enhances the reasoning capacities of large language models, was originally pioneered by Wei et al. (2022). The approach focuses on augmenting few-shot examples with detailed reasoning steps, thereby markedly improving performance on reasoning tasks. Subsequent works, inspired by Wei et al. (2022), have further refined the CoT methodology, such as Self-Consistency (Wang et al., 2022), Least-to-Most prompting (Zhou et al., 2022b), Dynamic Least-to-Most prompting (Drozdov et al., 2022), Self-Training (Huang et al., 2022), Verifier (Li et al., 2022) and Tree of Thought (Yao et al., 2023). The aforementioned methods primarily focus on improving the specific format of CoT prompting to better stimulate the reasoning capabilities of LLMs ($\geq$100B). However, they are not tailored to augment the reasoning capabilities of SLMs ($\leq$ 10B). We propose a novel method specifically designed to enhance the performance of SLMs on reasoning tasks.

### 2.2 Reasoning Enhancement in SLMs

Chung et al. (2022) observes that training SLMs with data that include intermediate reasoning steps can improve the reasoning capabilities of SLMs. Both Magister et al. (2022) and Ho et al. (2022) enhance the reasoning capabilities of SLMs by

fine-tuning them with training data, which includes intermediate reasoning steps generated by LLMs. STaR (Zelikman et al., 2022) enables the model to self-improve through its own generated rationales. SpecialFT (Fu et al., 2023) employs LLMs as teacher models and utilizes distribution matching in knowledge distillation to transfer the reasoning capabilities from LLMs to SLMs. Orca (Mukherjee et al., 2023) learns to imitate the reasoning process of LLMs from rich signals generated by LLMs, including explanation traces, step-by-step thought processes and other complex instructions. Differently, DialCoT transforms solving complex reasoning questions into decomposing questions and addressing a series of simpler questions, significantly reducing the task difficulty. Furthermore, we incorporate the PPO algorithm to enable the model to choose the optimal reasoning path among multiple options, thereby further enhancing the performance in reasoning tasks. Notably, our method does not require generating a large amount of training data with intermediate reasoning steps through LLMs. For example, by simply fine-tuning with only 7,000 examples from the GSM8K dataset, we could achieve a remarkable enhancement in SLM performance on reasoning tasks.

## 2.3 Question Decomposition

Question decomposition is crucial for understanding and solving complex questions. Earlier research (Kalyanpur et al., 2012) uses decomposition rules based on lexico-syntactic features to facilitate question decomposition. HSP (Zhang et al., 2019) proposes a hierarchical semantic parsing method based on a sequence-to-sequence model, which combines a question decomposer and an information extractor. Patel et al. (2022) designs a human-in-the-loop question decomposition method to improve model performance. Least-to-Most prompting (Zhou et al., 2022b) improves the format of CoT, enhancing the reasoning capabilities of LLMs by decomposing problems. Self-Ask (Press et al., 2022) explicitly asks itself follow-up questions before answering the initial question to perform compositional reasoning tasks. DecomT5 (Zhou et al., 2022a) develops robust decomposition-based models using distant supervision from comparable texts. Decomposition Distillation (Shridhar et al., 2022) learns a semantic decomposition of the original question into a sequence of sub-questions and uses it to train two models designated for question decomposition and resolution. Compared to the aforementioned methods, we not only decompose the question but also enable the model to choose the optimal reasoning path through reinforcement learning methods, thereby further enhancing the model's capability to solve complex questions.

## 3 Dialogue-guided Chain-of-Thought

We propose Dialogue-guided Chain-of-Thought (DialCoT), which aims to decompose complex questions into sub-questions in a dialogue format and gradually guide the model to obtain the final answer. Specifically, we introduce two roles for the model, namely the *Decomposer* and the *Solver*, who engage in a dialogue-based interaction. The Decomposer is responsible for breaking down the original question into a series of simpler sub-questions, while the Solver sequentially answers these sub-questions. We design distinct instructions for the Decomposer and Solver, followed by performing instruction tuning (Wei et al., 2021) on SLMs. We first introduce three different forms of DialCoT. Subsequently, we describe how we incorporate the Proximal Policy Optimization (PPO) algorithm into DialCoT to enable the model to select the optimal reasoning path and further enhance its reasoning capabilities.

### 3.1 Three Forms of DialCoT

We propose three different dialogue forms of DialCoT, namely DialCoT-A, DialCoT-M, and DialCoT-S. Specifically, DialCoT-A aims to guide SLMs in reasoning through minimal dialogue turns. DialCoT-M refines the Solver based on DialCoT-A, further reducing the task complexity. DialCoT-S maximally decomposes intermediate reasoning steps, allowing it to reference previous sub-questions and their answers when proposing new sub-questions. Figure 2 and Figure 3(1) presents the overall frameworks of them.[2] Next, we will introduce each of these forms individually.

**DialCoT-A (All at once).** We first establish an instruction for the Decomposer, enabling it to generate all sub-questions in a single step. Subsequently, we incorporate the generated sub-questions to the original texts and design a new instruction for the Solver, which allows the Solver to answer all sub-questions simultaneously. Figure 2(1) displays

---

[2]The specific prompt structures can be found in Table 3 of Appendix A.

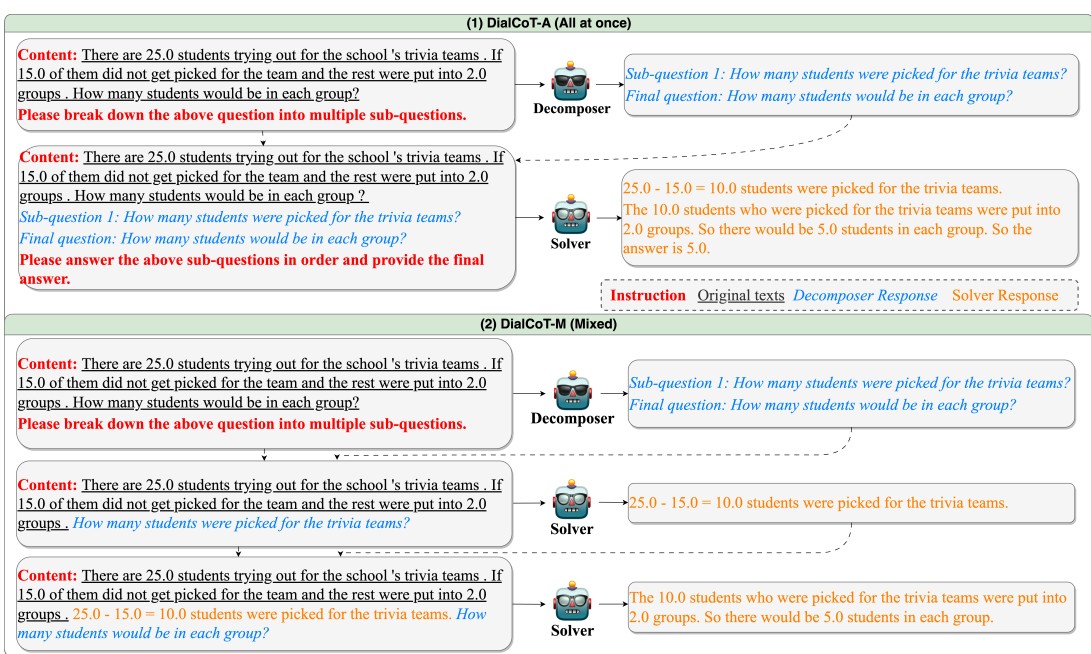

Figure 2: (1) **DialCoT-A**, where the Decomposer generates all sub-questions at once, and the Solver responds to all the sub-questions in a single step. (2) **DialCoT-M**, where the Decomposer is the same as in DialCoT-A, while the Solver addresses a sub-question at a single step, with the response being incorporated into the original texts to aid in solving subsequent sub-questions.

the instructions we designed and examples of input/output when the model operates as a Decomposer and Solver. DialCoT-A shares a similar motivation with Orca (Mukherjee et al., 2023), both striving to improve model reasoning performance by providing explicit reasoning paths. Orca represents the reasoning path through problem-solving steps, whereas our method exhibits the reasoning path via a sequence of sub-questions.

**DialCoT-M (Mixed).**   Upon deriving a series of sub-questions via the same Decomposer used in DialCoT-A, we sequentially replace the final question in the original texts with these sub-questions, which allows the Solver to address each sub-question individually. The Solver's response from each sub-question is appended to the original text, providing contextual support for solving subsequent questions. Figure 2(2) presents an example of DialCoT-M solving a math word problem. Compared to DialCoT-A, DialCoT-M mitigates the task complexity for the Solver by addressing a single and simpler question in each step.

**DialCoT-S (Step by Step).**   We design new instructions to direct the Decomposer to generate only a single sub-question at a step and the Solver to address the sub-question. Responses are prefixed with role identifiers such as "*Decomposer:* " and

"*Solver:* ". The history of the dialogue is appended after the original texts, aiding the model in answering subsequent questions and deriving the final answer. Figure 3(1) displays the overall framework of DialCoT-S. Compared to the previous two forms of DialCoT, DialCoT-S can reference previous sub-questions and their answers when generating new sub-questions. Moreover, DialCoT-S is more similar to the traditional multi-turn dialogue format. Therefore, it can more effectively stimulate the model's multi-turn dialogue capability to improve the model performance on reasoning tasks.

## 3.2   DialCoT-S-PPO

DialCoT-S-PPO aims to enable the model to select the optimal reasoning path by combining DialCoT-S with the PPO algorithm, further improving the model's performance on reasoning tasks. Figure 3(2) presents an example of DialCoT-S-PPO solving a math word problem. DialCoT-S-PPO chooses the optimal intermediate questions or answers from the model's multiple outputs, thus forming an reasoning path through a series of choices. Specifically, we first need to collect some data composed of states $\mathcal{S}$, actions $\mathcal{A}$ and rewards $\mathcal{R}$ for training the policy network $\pi_\theta$. $\mathcal{S}$ represents the space of states of the environment, which are the input of the policy network. Let $\mathbf{s}_t \in \mathcal{S}$ be a state

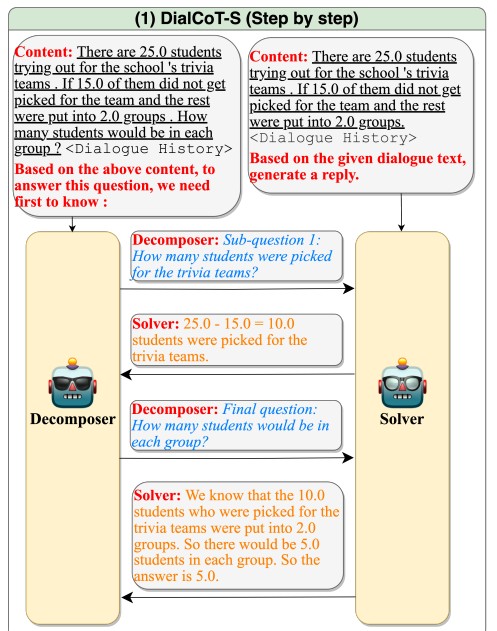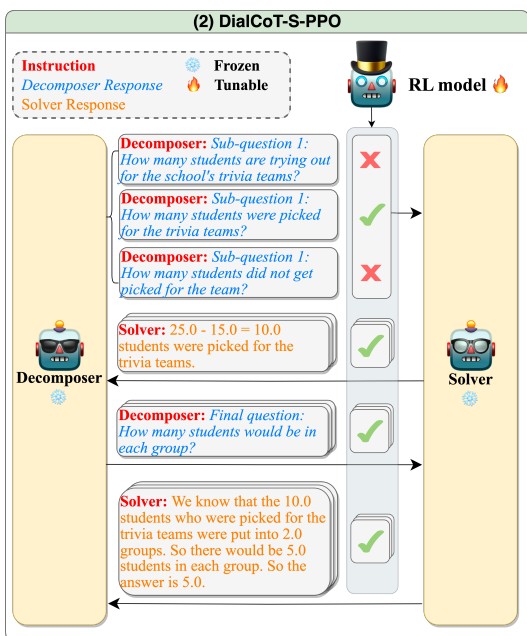

Figure 3: (1) **DialCoT-S**, where the Decomposer presents a sub-question at a step and the Solver answers it. Their past dialogue information is inserted into `<Dialogue History>` to assist in generating the answer of the final question. (2) An example of **DialCoT-S-PPO** solving a math word problem. The policy network is used to select a response from each step of the SLM, ultimately forming an optimal reasoning path and arriving at the final answer.

at time $t$ defined as

$$\mathbf{s}_t = [\mathbf{h}_1; \mathbf{h}_2; ...; \mathbf{h}_k], \quad (1)$$

$$[\mathbf{h}_1, \mathbf{h}_2, ..., \mathbf{h}_k] = \text{LM}_\phi(X), \quad (2)$$

where $X$ denotes the input text constructed via DialCoT-S, $\text{LM}_\phi(\cdot)$ represents the SLM after instruction tuning and $\mathbf{h}$ denotes the last hidden state of the model's response. We utilize beam search to generate the top-$k$ responses with the highest probability as candidates and $\mathbf{s}_t$ is the concatenation of these candidates' last hidden states. $\mathcal{A} = [0, 1, ..., k]$ represents the action space. At each time $t$, we input $\mathbf{s}_t$ into $\pi_\theta$ to obtain the probability $\mathbf{p}$ of actions:

$$\mathbf{p} = \pi_\theta(\mathbf{s}_t). \quad (3)$$

Based on the probability $\mathbf{p}$, $\pi_\theta$ chooses an action $\mathbf{a}_t \in \mathcal{A}$, which represents choosing the $\mathbf{a}_t$-th candidate. During the exploration phase, $\pi_\theta$ obtains $\mathbf{a}_t$ through sampling. In the inference phase, $\pi_\theta$ chooses the $a_t$ with the highest probability. When the model correctly answers a sub-question, it receives a reward $r_m \in [0, 1]$. $r_m$ is a hyperparameter that represents the extent to which the model focuses on the intermediate steps. When the model correctly answers the final question, it receives a reward $r_f = 1$. In all other cases, the reward is

0. We update the policy network $\pi_\theta$ through the following objective function:

$$J(\theta) = \mathbb{E}_\pi \left[ \min\left( \frac{\pi_\theta(\mathbf{a}_t \mid \mathbf{s}_t)}{\pi_{\theta_{\text{old}}}(\mathbf{a}_t \mid \mathbf{s}_t)} A_t, \right. \right.$$
$$\left. \left. \text{clip}\left( \frac{\pi_\theta(\mathbf{a}_t \mid \mathbf{s}_t)}{\pi_{\theta_{\text{old}}}(\mathbf{a}_t \mid \mathbf{s}_t)}, 1 - \epsilon, 1 + \epsilon \right) A_t \right) \right], \quad (4)$$

where $\pi_\theta(\cdot)$ represents the training policy network and $\pi_{\theta_{\text{old}}}$ denotes the policy network that interacted with the environment to collect data. Further information regarding the $\text{clip}(\cdot)$ and $A_t$ can be found in Schulman et al. (2017). After update the parameters of $\pi_\theta$, the new parameter of $\pi_\theta$ is transmitted to $\pi_{\theta_{\text{old}}}$. We then repeat data collection and $\pi_\theta$ updates until training completion.

## 4 Experiments

### 4.1 Datasets

We consider the following four math word problem datasets: **GSM8K** (Cobbe et al., 2021), **Multi-Arith** (Roy and Roth, 2015), **ASDiv** (Miao et al., 2020) and **SVAMP** (Patel et al., 2021). The GSM8K dataset contains 7,000 training instances with intermediate questions and answers. In contrast to previous work (Magister et al., 2022; Ho et al., 2022; Fu et al., 2023), we only fine-tune

our model using these 7,000 instances, eliminating the need for generating additional training data with intermediate reasoning steps via LLMs. Apart from evaluating our method on the GSM8K test set, we evaluate the model's out-of-distribution performance on three other datasets. All datasets comprise arithmetic reasoning problems at a primary school level, varying by the entities they incorporate. This form of out-of-distribution generalization is typically classified as lexical-level compositional generalization (Liu et al., 2021). Following SpecialFT (Fu et al., 2023), for each dataset, we employ 500 instances as the validation set, using the remaining instances (800 for GSM8K, 400 for MultiArith, 18K for ASDiv, 500 for SVAMP) as the test set.

## 4.2 Baselines

In our experiments, we compare our method with some competitive baselines which can be grouped into two categories: (1) *generic large language models*: **code-davinci-002** (Chen et al., 2021) presumably with a size of 175B or more, **LaMDA-137B** (Thoppilan et al., 2022), **PaLM-60B** (Chowdhery et al., 2022) and **UL2-20B** (Tay et al., 2022), each of which exhibits strong reasoning abilities in Chain-of-Thought prompting. (2) *concurrent works enhancing SLMs' reasoning capabilities*: **CoT-FT** (Wei et al., 2021) directly employs the 7000 CoT training instances from the GSM8K dataset to perform instruction tuning, which is a vanilla approach to enhancing the reasoning capabilities of SLMs. **DecomDistill** (Shridhar et al., 2022) is a decomposition-based method, which learns a semantic decomposition of the original problem into a sequence of sub-problems through LLMs. Both Magister et al. (2022) and Ho et al. (2022) fine-tune SLMs by generating training data with intermediate reasoning steps through LLMs. **SpecialFT** (Fu et al., 2023) employs LLMs as teacher models and utilizes distribution matching in knowledge distillation to transfer the reasoning capabilities from LLMs to SLMs. It is noted that SpecialFT uses 130K training instances with intermediate reasoning steps generated by LLMs, which is nearly twenty times the size of our training set.

## 4.3 Implementation

We consider using FlanT5-XL (3B)/XXL (11B) as the backbone of our model. The Decomposer and Solver utilize different instructions (as shown in Figure 2) but share the same model parameters.

| Hyperparameters | Scope |
|---|---|
| learning rate | $\{1e-4, \mathbf{3e\text{-}4}, 5e-4, 1e-3\}$ |
| batch size | $\{1024, 2048, \mathbf{4096}\}$ |
| $\epsilon$ | $\{0.1, \mathbf{0.2}, 0.3\}$ |
| $k$ | $\{2, \mathbf{3}, 4, 5, 6\}$ |
| $r_m$ | $\{0.1, 0.2, \mathbf{0.3}, 0.4, 0.5\}$ |

Table 1: The searching scope for the hyperparameters of the proximal policy optimization algorithm. We highlight the best settings in bold.

Following Chung et al. (2022), we fine-tune the model for 50 epochs with the batch size 4096 and the learning rate $5e-4$. For the PPO algorithm, we use three feed-forward layers as the policy network and set the number of hidden units to 1024. Moreover, we use the grid search to find the best hyperparameters. The details of grid search are shown in Table 1. As a result, we set the learning rate as $3e-4$ and batch size as 4096 for the policy network. We also set $\epsilon$ to 0.2, $k$ to 3 and $r_m$ to 0.3. During the stage of optimizing the policy network, we freeze the backbone parameters. All baseline results except CoT-FT (Wei et al., 2021) are recorded in SpecialFT (Fu et al., 2023). For CoT-FT and our method, we keep the experimental setup consistent with other baselines. We run all the experiments on eight NVIDIA Tesla A100 GPU.

## 4.4 Results

Table 2 shows the performance of various methods on four arithmetic reasoning datasets. First, we discuss the results for DialCoT-A, DialCoT-M, and DialCoT-S. Then, we compare our method with other baselines to demonstrate its superiority. Finally, we validate the effectiveness of incorporating the PPO algorithm based on DialCoT-S through an ablation study.

**Discussion of Three DialCoT Variants.** Compared to CoT-FT, all three forms of DialCoT outperform on the four reasoning tasks, demonstrating their effectiveness in enhancing the reasoning capacities of SLMs. More specifically, DialCoT-M performs better than DialCoT-A. This indicates that SLMs lack the capability to decompose a reasoning problem and answer it all at once. DialCoT-M addresses only one sub-question at a single step, which reduces the task difficulty and makes it more suitable for SLMs. DialCoT-S, in comparison to DialCoT-M, shows greater performance gains, which can be attributed to two factors: (1)

| Methods | Backbone | #Params. | GSM8K | MultiArith | ASDiv | SVAMP | Average |
|---|---|---|---|---|---|---|---|
| *Generic Large Language Models* | | | | | | | |
| code-davinci-002 (Chen et al., 2021) | | $\geq$ 175B | 63.1 | 95.8 | 80.4 | 76.4 | 78.9 |
| LaMDA (Thoppilan et al., 2022) | | 137B | 14.8 | 45.0 | 46.6 | 37.5 | 36.0 |
| PaLM (Chowdhery et al., 2022) | | 60B | 29.9 | 75.0 | 61.9 | 46.7 | 53.2 |
| UL2 (Tay et al., 2022) | | 20B | 4.4 | – | 16.9 | 12.5 | – |
| *Concurrent Works to Boosting SLM Reasoning* | | | | | | | |
| DecomDistill[†](Shridhar et al., 2022) | GPT | 7B | 21.0 | – | – | – | – |
| Magister et al. (2022)[†] | T5-XXL | 11B | 21.9 | – | 42.1 | – | – |
| Ho et al. (2022)[†] | GPT | 6B | 6.8 | 33.3 | – | – | – |
| CoT-FT (Wei et al., 2021) | FlanT5-XL | 3B | 13.5 | 24.0 | 20.7 | 17.7 | 19.0 |
| SpecialFT[†] (Fu et al., 2023) | FlanT5-XL | 3B | 22.4 | 42.3 | 28.4 | 23.8 | 29.3 |
| DialCoT-A (All at once) | FlanT5-XL | 3B | 20.3 | 40.3 | 24.6 | 21.3 | 26.6 |
| DialCoT-M (Mixed) | FlanT5-XL | 3B | 22.9 | 43.1 | 27.1 | 23.2 | 29.1 |
| DialCoT-S (Step by Step) | FlanT5-XL | 3B | 24.3 | 45.7 | 29.3 | 25.5 | 31.2 |
| DialCoT-S-PPO | FlanT5-XL | 3B | 25.6 | 46.9 | 30.7 | 27.1 | 32.6 |
| CoT-FT (Wei et al., 2021) | FlanT5-XXL | 11B | 16.1 | 51.7 | 36.5 | 39.7 | 36.0 |
| SpecialFT[†] (Fu et al., 2023) | FlanT5-XXL | 11B | 27.1 | 63.0 | 37.6 | 35.6 | 40.8 |
| DialCoT-A (All at once) | FlanT5-XXL | 11B | 21.7 | 57.1 | 32.5 | 34.2 | 36.4 |
| DialCoT-M (Mixed) | FlanT5-XXL | 11B | 30.5 | 63.9 | 38.2 | 37.7 | 42.6 |
| DialCoT-S (Step by Step) | FlanT5-XXL | 11B | 35.2 | 65.7 | 39.3 | 40.3 | 45.1 |
| **DialCoT-S-PPO** | FlanT5-XXL | 11B | **37.1** | **68.1** | **40.9** | **41.7** | **47.0** |

Table 2: Accuracy (%) of various methods on four reasoning tasks. [†] indicates that the method employs additional training data with intermediate reasoning steps generated via LLMs, where SpecialFT uses nearly 20 times the training data of our method. We highlight the best results on SLMs ($\sim$10B) in bold.

DialCoT-S obtains more intermediate information before generating sub-questions. (2) DialCoT-S more effectively stimulate the model's multi-turn dialogue capabilities to boost its reasoning performance.[3]

**Comparison between DialCoT and Baselines.** From the Table 2, we observe that DialCoT-S-PPO attains state-of-the-art results on SLMs. Specifically, when using FlanT5-XXL as the backbone, DialCoT-S-PPO improves the average performance across the four datasets by 6.2% compared to SpecialFT. Notably, the training data we used is only 1/20 of what SpecialFT used, which clearly demonstrates that our method is very effective in improving reasoning capabilities of SLMs. On the other hand, when compared with LLMs, all variations of DialCoT (i.e., DialCoT-A/M/S/PPO) outperform LaMDA-137B on average across the four datasets, despite the parameters of our approach are merely 1/12 of LaMDA's. This further substantiates the superiority of our approach. While there is still a noticeable gap when compared to code-davinci-002, our experimental results demonstrate that there is potential for SLMs to achieve LLM-level reasoning

capabilities via appropriate fine-tuning methods.[4]

**Ablations.** The ablation study is conducted to demonstrate the effectiveness of incorporating the PPO algorithm based on DialCoT-S. Compared to DialCoT-S with FlanT5-XXL, DialCoT-S-PPO achieves an improvement of nearly 2%, confirming the effectiveness of employing the PPO algorithm for selecting the optimal reasoning path. Additionally, we observe that when using FlanT5-XL as the backbone, the performance gain brought by the PPO algorithm is 1.4%, which is lower than the performance on FlanT5-XXL. This might be attributed to the lower diversity in the multiple replies generated by the smaller model.

### 4.5 Analysis

**Different Model Size.** We extend our method to smaller backbones, including FlanT5-Base (250M) and FlanT5-Large (760M), on the GSM8K and MultiArith datasets. Our experimental results are illustrated in Figure 4. In comparison to the original FlanT5, our approach improves the performance of the model on reasoning tasks across different model sizes, affirming the effectiveness of our method for varying model sizes. Notably, we observe that

---

[3]A more detailed discussion of the three DialCoT variants can be found in Appendix D.

[4]The detailed experimental comparison between DialCoT and SelfAsk can be found in Appendix A.

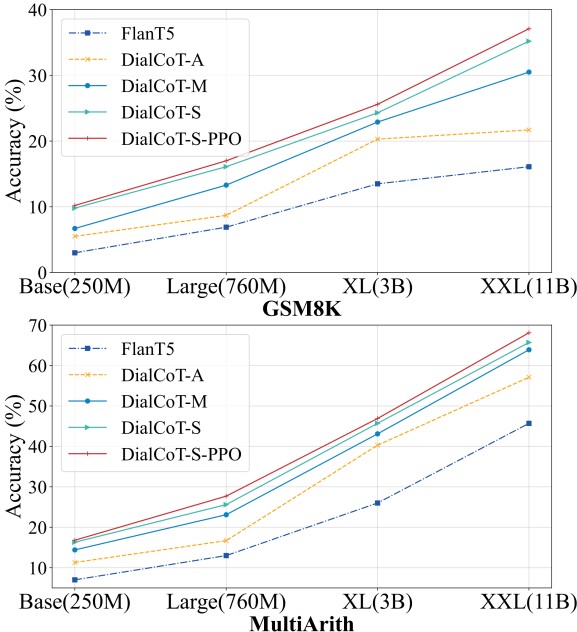

Figure 4: Results under different model sizes on the GSM8K and MultiArith datasets. FlanT5 indicates the results of using Few-shot CoT (Wei et al., 2022) on the backbone (Chung et al., 2022). Our methods achieve performance improvements across all model sizes.

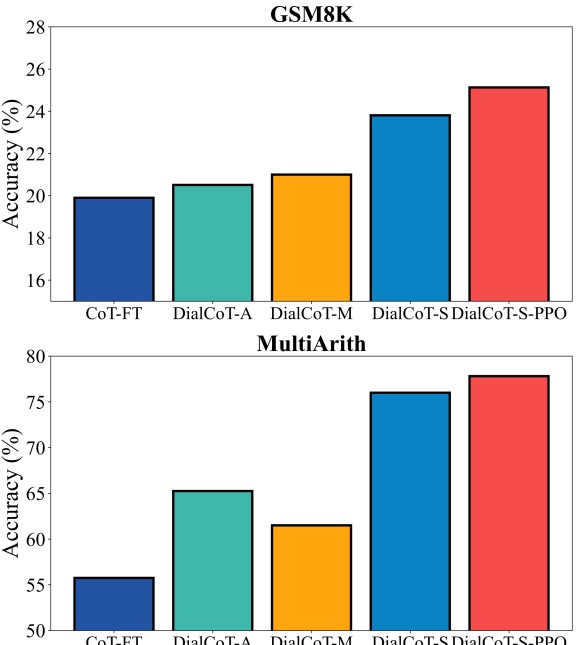

Figure 5: Results using LLaMA-7B (Touvron et al., 2023) as the backbone on the GSM8K and MultiArith datasets. Our method achieves significant performance gains on decoder-only LMs.

our method yields larger performance gains on larger model sizes, which is similar to the results of Chung et al. (2022). This could be attributed to the stronger capabilities that larger models obtain during pre-training, making them more readily stimulated.

**Different Model Architectures.** To evaluate the generalizability of DialCoT across LMs with varying architectures, in addition to the encoder-decoder LM (e.g., FlanT5), we conduct experiments using the decoder-only LM (e.g., LLaMA-7B (Touvron et al., 2023)) as the backbone of our method on the GSM8K and MultiArith datasets. The results are illustrated in Figure 5. As can be seen from the figure, all of our methods achieve significant performance gains compared to CoT-FT, especially DialCoT-S and DialCoT-S-PPO. This demonstrates that our approach is applicable to SLMs with various architectures, not merely effective on encoder-decoder LMs. Moreover, we observe that DialCoT-A performs better than DialCoT-M on the MultiArith dataset, which is different from the results based on FlanT5 (as shown in Figure 4). This suggests that the most suitable form of DialCoT may differ for different SLMs, which could potentially be related to model architecture and pre-training corpora.

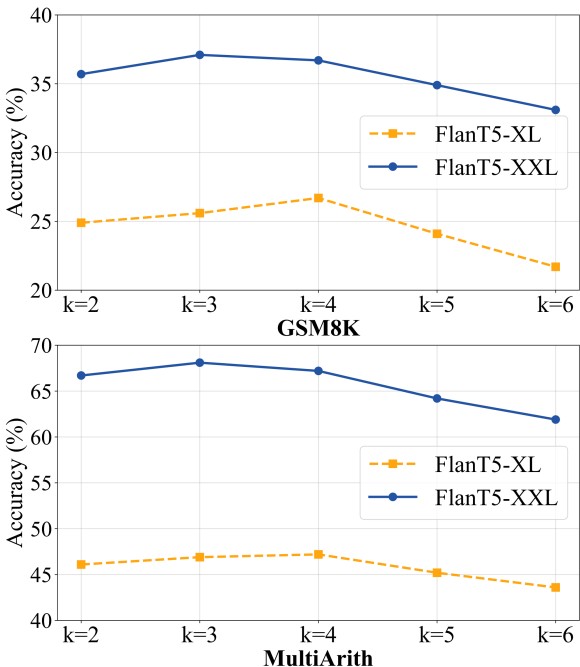

Figure 6: The effect of hyperparameter $k$ on the model performance on the GSM8K and MultiArith datasets.

**Effect of Hyperparameter $k$.** We set the model to output the top-$k$ responses with the highest probability in each step of dialogue through beam search. In other words, $k$ represents the size of the action space, indicating that we can select the optimal response from $k$ different responses in each

step. Figure 6 illustrates the effect of $k$ on the model performance on the GSM8K and MultiArith datasets. Specifically, the model can achieve optimal performance when $k$ is set to 3 or 4. When $k$ is too small, the number of reasoning paths we can choose from is too limited, preventing us from achieving optimal performance. Conversely, when $k$ is too large, the space of available reasoning paths is too large, which may introduce noise and make it difficult for the model to learn to select the optimal reasoning path.

# 5 Conclusion

In this paper, we explored strategies to boost the reasoning capabilities of SLMs and proposed Dial-CoT, which aims to generate intermediate reasoning steps in a dialogue format leading to the final answer. Specifically, we designed two roles for the model, namely Decomposer and Solver. The Decomposer is responsible for breaking down questions into multiple sub-questions, while the Solver is tasked to address the sub-questions. They engage in dialogue to arrive at the final answer. We introduced three different dialogue formats: DialCoT-A (All at once), DialCoT-M (Mixed) and DialCoT-S (Step by step). Furthermore, we incorporated the PPO algorithm into DialCoT-S to enable the model to choose the optimal reasoning path among multiple options, thereby further improving its performance on reasoning tasks. We conducted extensive experiments on four arithmetic reasoning datasets and the experimental results demonstrate the effectiveness of our method. Future work primarily involves extending our method to other types of reasoning tasks, such as commonsense reasoning and symbolic reasoning. In addition, we will explore other decomposition methods or other reinforcement learning methods to optimize the reasoning paths of SLMs.

# Limitations

We conduct experiments on four arithmetic reasoning tasks, demonstrating the effectiveness of Dial-CoT. However, as our reward pattern is specifically designed for arithmetic reasoning, modifications are necessary to apply our method to commonsense or symbolic reasoning. This presents a limitation to the broader applicability of our method. We plan to extend DialCoT to a wider range of reasoning tasks in the future. On the other hand, DialCoT specifically focuses on enhancing the reasoning

capabilities of SLMs. Due to constraints on computational resources, we do not conduct experiments on larger scale language models ($\geq$ 20B), thus the applicability of our method for LLMs remains undetermined. We will further explore the performance of DialCoT on larger scale language models in future research.

# Ethics Statement

The proposed method has no obvious potential risks. All the scientific artifacts used/created are properly cited/licensed, and the usage is consistent with their intended use. Also, we open up our codes and hyper-parameters to facilitate future reproduction without repeated energy cost.

# Acknowledgements

This work has been supported by the National Natural Science Foundation of China under Grant No.U1911203, the National Natural Science Foundation of China under Grant No.62377012 and Fundamental Research Funds for the Central Universities under grant number YBNLTS2023-015.

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

## A Comparison between DialCoT and SelfAsk

We add further discussion regarding the comparison with SelfAsk. Self-Ask (Press et al., 2022) explicitly asks itself follow-up questions before answering the initial question to perform compositional reasoning tasks. Please refer Table 3 for the comparison between SelfAsk and our methods.

From the table, we see that: (1) SelfAsk is designed for in-context learning method without fine-tuning, whereas DialCoT is a finetune-based method. (2) Even if fine-tuning can be applied to SelfAsk theoretically, how to format the fine-tuning is still un-explored. We offer a novel way to leverage two tailored tasks (problem decomposition and problem solving) through fine-tuning task-oriented instructions on the same model (SLM). Therefore, we believe the performance gain mainly come from the way of decomposing and solving sub-questions through fine-tuned model with tailored instructions. The additional instructions naturally come along with the proposed solution. They will instruct the model to play dedicated role during the instruction fine-tuning. However, we don't think they are the major factors.

Moreover, we conduct additional experiments (SelfAsk) on the GSM8K dataset using Flan-T5-XXL. The results are illustrated in Table 4. From the table, we can draw the following conclusions:

Firstly, in the fine-tuning setting, SelfAsk improves on Standard CoT by 5.5% with all-at-once finetuning and by 13.2% with sequential finetuning. We believe these improvements stem from problem decomposition. Furthermore, SelfAsk with sequential finetuning improves by 7.7% compared to SelfAsk with all-at-once finetuning. This once again confirms our previous conclusion that decomposing problems sequentially is more effective than decomposing them all at once. Moreover, DialCoT-S improves by 5.9% over SelfAsk with sequential finetuning. We attribute this additional improvement to fine-tuning with different instructions tailored for specific tasks. Compared to SelfAsk with sequential finetuning, DialCoT-S has clearer and more independent instructions for both problem decomposition and problem solving.

Secondly, compared to finetune-based methods, methods without fine-tuning perform poorly, indicating that fine-tuning is crucial for SLMs. At the same time, we found that both SelfAsk and DialCoT-S experience a performance drop in the setting compared to Standard CoT. This could be either because Flan-T5 was trained with some standard-CoT-formatted training data or due to the weaker instruction-following capabilities of SLMs, where complex instructions increase the task difficulty.

For encoder-decoder structure, i.e., T5, the difference between SelfAsk-A and SelfAsk-S is significant, due to the fact of bidirectional attention within the input. For decoder-only structure, the difference is indeed very subtle. Table 6 shows the results of additional experiments (SelfAsk) on the GSM8K using LLaMA-7B. The SelfAsk-A outputs all intermediate questions, answers and connecting words between them such as "Follow up" and "Intermediate answer", while the sequential finetuning is focused on outputting intermediate question or answer, and does not include the loss of connecting words, encouraging the model to focus more on most important part of the learning. We speculate this subtlety brings the improvement.

## B Fine-grained Analysis on Each Sub-step

We select samples that require decomposition into three sub-questions from all test sets and report the accuracy for each sub-question in Table 7. As shown in the table, as the number of steps increases, i.e., as the complexity of the questions rises, our method shows a notable performance improvement (7%) compared to CoT-FT. On the other hand, although our method's performance does decline as question complexity increases, the rate of decline is significantly slower compared to vanilla CoT. When comparing DialCoT-S with DialCoT-S-PPO, It is evident that step-level PPO significantly improves the model's performance on reasoning tasks at every step.

## C Comparison of Inference Speed

In terms of the time cost for reasoning, the inference speed of our method is comparable to that of SelfAsk (Press et al., 2022), as both need to generate sub-questions and answers. In terms of the number of iterations, DialCoT-A requires two iterations, which is comparable to Zero-shot CoT (Kojima et al., 2022), while DialCoT-M and DialCoT-S require approximately three and six iterations respectively. Overall, in our methods, DialCoT-A has the fastest inference speed, followed by DialCoT-M, and finally DialCoT-S. The trade-off between

| Methods | Prompt Structure | All-at-once? | ICL or FT |
|---------|------------------|--------------|-----------|
| SelfAsk | *Input:* Original question | *Output:* Sub-question 1 + Intermediate step 1 + ... + Final answer | All-at-once | In-context Learning |
| DialCoT-A | **Decomposer:** *Input:* Original question | *Output:* Sub-questions
**Solver:** *Input:* Original question + Sub-questions | *Output:* Intermediate steps + Final answer | All-at-once | Fine-tuning |
| DialCoT-M | **Decomposer:** *Input:* Original question | *Output:* Sub-questions
**Solver:** *Input:* Original question + Sub-question 1 | *Output:* Intermediate step 1
**Solver:** *Input:* Original question + Intermediate step 1 + Sub-question 2 | *Output:* Intermediate step 2
...
**Solver:** *Input:* Original question + Intermediate steps + Final question | *Output:* Final answer | Output intermediate steps sequentially | Fine-tuning |
| DialCoT-S | **Decomposer:** *Input:* Original question | *Output:* Sub-question 1
**Solver:** *Input:* Original question + Sub-question 1 | *Output:* Intermediate step 1
**Decomposer:** *Input:* Original question + Sub-question 1 + Intermediate step 1 | *Output:* Sub-question 2
**Solver:** *Input:* Original question + Sub-question 1 + Intermediate step 1 + Sub-question 2 | *Output:* Intermediate step 2
...
**Decomposer:** *Input:* Original question + Sub-questions + Intermediate steps | *Output:* Final question
**Solver:** *Input:* Original question + Sub-questions + Intermediate steps + Final question | *Output:* Final answer | Output sub-questions and intermediate steps alternately | Fine-tuning |

Table 3: Detailed Comparison between DialCoT and SelfAsk. The table omits the specific details of the prompts and instead focuses on the input-output formats of the prompts. DialCoT is fine-tuned using distinct instructions tailored for two specialized tasks (Decomposer and Solver) on a shared model.

| Method | Finetune or not | GSM8K |
|--------|-----------------|-------|
| Standard CoT | finetune | 16.1 |
| SelfAsk | all at once, finetune | 21.6 |
| SelfAsk | sequencially, finetune | 29.3 |
| DialCoT-S | finetune | 35.2 |
| Standard CoT | without finetune | 12.7 |
| SelfAsk | without finetune | 11.3 |
| DialCoT-S | without finetune | 10.9 |

Table 4: Accuracy (%) of various methods on the GSM8K dataset. The fine-tuning dataset used for all experiments is the GSM8K training set. The prompt structures corresponding to the two different fine-tuning methods for SelfAsk are shown in Table 5.

inference time and performance needs to be considered. If faster inference speed is required, one can opt for DialCoT-A, at the expense of some performance loss. Conversely, if better performance is the priority, DialCoT-S can be chosen, although this would come at the cost of increased inference time.

## D   Detailed Discussion of Three DialCoT Variants

From Table 3, we can clearly see the differences in the prompt structures of different methods. Combined with the experimental results from Table 2, we can draw the following conclusions: (1) In a side-by-side comparison between standard CoT and DialCoT-A, it becomes evident that DialCoT-A employs self-generated sub-problems as a strategic guide for formulating intermediate steps and solutions. The enhanced performance of DialCoT-A over standard CoT implies that self-generated navigation through these sub-problems can significantly enhance the reasoning capability of Smaller Language Models (SLMs). (2) When comparing DialCoT-M and DialCoT-A, the former opts for a sequential approach to answering sub-questions, as opposed to addressing them all-at-once. The superior performance metrics of DialCoT-M in comparison to DialCoT-A substantiate the claim that a sequential methodology for answering sub-questions yields greater efficacy than an all-at-once approach. (3) The primary difference between DialCoT-S and DialCoT-M is that DialCoT-S generates sub-

| Method | Fine-tuning Method | Prompt Structure |
|---|---|---|
| SelfAsk | all at once | *Input:* Original question \| *Output:* **Follow up:** Sub-question 1 |
| | | **Intermediate answer:** Intermediate step 1 |
| | | **Follow up:** Sub-question 2 |
| | | **Intermediate answer:** Intermediate step 2 + ... + **Follow up:** Final question |
| | | **Intermediate answer:** Final answer |
| SelfAsk | sequencially | *Input:* Original question |
| | | **Follow up:** \| *Output:* Sub-question 1 |
| | | *Input:* Original question |
| | | Follow up: Sub-question 1 |
| | | **Intermediate answer:** \| *Output:* Intermediate step 1 |
| | | *Input:* Original question |
| | | Follow up: Sub-question 1 |
| | | Intermediate answer: Intermediate step 1 |
| | | **Follow up:** \| *Output:* Sub-question 2 |
| | | ... |
| | | *Input:* Original question |
| | | Follow up: Sub-question 1 |
| | | Intermediate answer: Intermediate step 1 + ... + **Follow up:** \| *Output:* Final question |
| | | *Input:* Original question |
| | | Follow up: Sub-question 1 |
| | | Intermediate answer: Intermediate step 1 + ... + Follow up: Final question |
| | | **Intermediate answer:** \| *Output:* Final answer |

Table 5: Prompt structures corresponding to the two different fine-tuning methods for SelfAsk.

| Method | GSM8K |
|---|---|
| SelfAsk + All-at-once finetuning | 21.00 |
| SelfAsk + Sequential finetuning | 22.95 |

Table 6: Accuracy (%) of SelfAsk using LLaMA-7B on the GSM8K dataset.

| Method | first step | second step | final step |
|---|---|---|---|
| CoT-FT | 55.2 | 41.9 | 34.7 |
| DialCoT-S | 60.7 | 48.8 | 43.8 |
| DialCoT-S-PPO | 63.9 | 51.2 | 45.3 |

Table 7: Accuracy (%) of various methods using FlanT5-XXL on the GSM8K dataset for each sub-step. CoT-FT refers to the results obtained by replacing the original question with different step-based sub-questions.

questions sequentially rather than generating all sub-questions at once. Given DialCoT-S's stronger performance metrics, this indicates that the approach of sequentially decomposing sub-questions is more effective than generating all sub-questions in a single step.