# OpenReview forum: "DialCoT Meets PPO: Decomposing and Exploring Reasoning Paths in Smaller Language Models"
_EMNLP/2023/Conference — EMNLP 2023 Main_

### Official Review · Reviewer_v8cq · 2023-07-24

**Soundness:** 4

**Excitement:**

3: Ambivalent: It has merits (e.g., it reports state-of-the-art results, the idea is nice), but there are key weaknesses (e.g., it describes incremental work), and it can significantly benefit from another round of revision. However, I won't object to accepting it if my co-reviewers champion it.

**Paper Topic And Main Contributions:**

The paper tackles the task of finetuning SLMs to perform symbolic reasoning tasks such as GSM8K. The proposed method is to format the data as dialog turns instead of regular CoT texts. The paper shows favorable performance of doing so on a variety of datasets. Furthermore, the incorporation of PPO further boosts accuracy by 1-2%.

**Questions For The Authors:**

Quite rarely, I don't have any minor questions except from the very important ones in Reasons To Reject. The paper is very well-written in this regard.

**Reasons To Accept:**

1. Assuming the findings of this paper, the dialog format has the potential to compete with or even replace the standard CoT format in certain scenarios.
2. The paper is well-presented with sound logic behind design choices.

**Reasons To Reject:**

My major objection to this paper's acceptance is:
1. The insufficiency of explaining why or how the dialog format is helpful. Dialog is a format. Regular CoT is also a format. Thus, it would be imperative to rigorously control for this variable by doing a precise side-by-side comparison with two finetuning datasets whose only difference is the format (dialog vs. regular CoT). The citing of CoT-FT (Wei et al., 2021) row does not sufficiently answer this question as there could be differences in the finetuning data. Even were I to be informed that CoT-FT (Wei et al., 2021) is exactly the same as the proposed work except the format, it is quite necessary to show what exactly is it in the format that boosts performance. Is it simply about using a question instead of a statement? Is it because punctuations like question marks (a little silly, but the point is made)?
Example of a strict side-by-side:
Dialog: Sub-question 1: How many students were picked for the trivia teams? Sover:...
CoT: First, I need to find out the number of students that were picked for the trivia teams. That is calculated as ...
Without stronger evidence that the **dialog format** is really the secret sauce here (a fact which is not that self-evident for LLMs), it's hard to gain insights from this paper beyond "this particular data format works well and I only have some speculations of why."

2. The usage of PPO seems to be a secondary contribution of the paper, but its effect is not sufficiently disentangled from that of the dialog format. The regular CoT format can also incorporate PPO. Without further experiments, we're left with a half-baked claim that "PPO works well with the dialog format" without understanding the whole picture.

3. The performance gain of DialCoT-A over CoT-FT, as well as that of DialCoT-S-PPO over DialCoT-S is rather small (considering literature that works on GSM8K), but I'll not use this point as a deal-breaker.

**Reproducibility:**

5: Could easily reproduce the results.

**Reviewer Confidence:**

5: Positive that my evaluation is correct. I read the paper very carefully and I am very familiar with related work.

---

> ### Author Rebuttal · Authors · 2023-08-29
>
> Before reviewing the responses, **please make sure to read the [general response](https://openreview.net/forum?id=VyjNXY2wgi&noteId=sjLPj6ptDw) first.** Thank you very much for your valuable comments. Below are our replies to the questions you have raised and we hope they can address your concerns.
>
> **R1:** "The insufficiency of explaining why or how the dialog format is helpful. Dialog is a format. Regular CoT is also a format.  ... Without stronger evidence that the dialog format is really the secret sauce here (a fact which is not that self-evident for LLMs), it's hard to gain insights from this paper beyond "this particular data format works well and I only have some speculations of why.""
>
> **A:** Firstly, **CoT-FT and our method use exactly the same training data** (7,000 samples from the GSM8K training set). The results in Table 2 confirm that **the form of DialCoT is superior to the regular form of CoT**. Secondly, by comparing the results of DialCoT-A and DialCoT-M, we find that **sequentially answering individual sub-questions is more effective than answering all sub-questions at once.** Comparing the results of DialCoT-M and DialCoT-S, we find that using a format to **sequentially break down the problem is more effective than decomposing all sub-questions at once.** The above are the insights that can be drawn from the experiments in this paper. For a more fine-grained analysis, such as the specific impact of punctuation, instructions, and other parts on model performance, we will conduct more experiments and analyses in the future. We hope our work could inspire more research efforts in this direction.
>
> **R2:** The usage of PPO seems to be a secondary contribution of the paper, but its effect is not sufficiently disentangled from that of the dialog format. The regular CoT format can also incorporate PPO. Without further experiments, we're left with a half-baked claim that "PPO works well with the dialog format" without understanding the whole picture.
>
> **A:** Firstly, we would like to clarify that our step-based PPO is not a secondary contribution but rather **one of the main contributions of this paper.** We are **the only ones to utilize the PPO method to optimize the intermediate reasoning steps of the SLMs.** This resulted in a further improvement on top of the already significant performance gains achieved by DialCoT-S. Specifically, our version of PPO is step-level, which is completely different from the token-level PPO in RLHF[1]. **Regular CoT cannot explicitly separate each reasoning step, so it's not suitable for the step-level PPO we propose.** In contrast, our method has effectively decomposed the reasoning steps, making it more suitable for step-level PPO.
>
> **R3:** The performance gain of DialCoT-A over CoT-FT, as well as that of DialCoT-S-PPO over DialCoT-S is rather small (considering literature that works on GSM8K), but I'll not use this point as a deal-breaker.
>
> **A:** Compared to DialCoT-M and DialCoT-S, DialCoT-A is most similar to regular CoT, and therefore yields smaller performance gains. As argued before, **our step-level PPO achieved an additional 2% improvement on top of the already substantial performance gains realized by DialCoT-S.** Considering that our test set contains 20,000 questions, this means an additional 400 questions were answered correctly, which is a stable and very noticeable performance boost.
>
> [1] Training language models to follow instructions with human feedback, NIPS 2022

---

### Official Review · Reviewer_orRw · 2023-07-31

**Soundness:** 3

**Excitement:**

4: Strong: This paper deepens the understanding of some phenomenon or lowers the barriers to an existing research direction.

**Paper Topic And Main Contributions:**

This paper uses multi-round dialogue to decompose the problem using small models and answer the decomposed sub-problems step by step. At the same time, the RL algorithm is used to select a better sub-problem decomposition/reasoning path, which improves the reasoning ability of small models.

**Questions For The Authors:**

QA) I doubt that this method has limitations due to multi-round dialogue. For example,  the paper does not show how many rounds of dialogue are usually required to solve a problem because this form of dialogue is equivalent to output sub-questions and answers at each step, which is longer than direct step-by-step reasoning. If the problem is more complex and the number of rounds of dialogue increases, will there be a significant decrease?

QB) Some parameters in the RL section are set relatively briefly. For example, why should rm be selected as 0.3 in Table 1 in the reward model? Is it differ from task types? The authors do not prove such a setting, which seems to be summarized from the experimental results of downstream tasks.

QC) Homogenization of task types. As the paper says, "All datasets comprise arithmetic reasoning problems at a primary school level, varying by the entities they incorporate", I doubt the generalization of methods on other tasks.

QD) How is the intermediate step data obtained in the training data, such as how to determine whether the solver answers correctly or incorrectly for a sub-problem decomposed from the decomposer?

QE) Should the action space of models of different sizes be different because the models have different abilities to decompose problems correctly?

QF) Is the method still effective on more complex datasets that require more reasoning steps and larger action space?

**Reasons To Accept:**

1. Introduce PPO to help the model choose a better reasoning path
2. Compared with the previous method, the performance has been greatly improved
3. The experimental part illustrates the effectiveness of the method to a certain extent through different settings

**Reasons To Reject:**

1. The Intro section explains two advantages, but the first advantage is the idea that CoT already exists, and the innovation is not enough to be the first contribution of the paper. Except for this part, the paper's overall innovation appears insufficient.
2. The purpose of using PPO is to choose a better reasoning path, but the experimental part only shows the overall effect after using the PPO method. It is recommended to supplement the model accuracy at different stages after dismantling sub-problems for different methods, which can better illustrate the method's effectiveness.
3. Using the dialogue method to reason, the comparison of reasoning speed and time cost compared with the previous method needs to be supplemented
4. Figure 2 is overcrowded, poor readability
5. Figure 3 is the main method diagram of the paper, which is separated from Figure 2, and it isn't easy to intuitively compare the differences between different methods

**Reproducibility:**

4: Could mostly reproduce the results, but there may be some variation because of sample variance or minor variations in their interpretation of the protocol or method.

**Reviewer Confidence:**

4: Quite sure. I tried to check the important points carefully. It's unlikely, though conceivable, that I missed something that should affect my ratings.

---

> ### Author Rebuttal · Authors · 2023-08-29
>
> Before reviewing the responses, **please make sure to read the [general response](https://openreview.net/forum?id=VyjNXY2wgi&noteId=sjLPj6ptDw) first.** Thank you very much for your valuable comments. Below are our replies to the questions you have raised and we hope they can address your concerns.
>
> **R1:** The Intro section explains two advantages, but the first advantage is the idea that CoT already exists, and the innovation is not enough to be the first contribution of the paper. Except for this part, the paper's overall innovation appears insufficient.
>
> **A:** Vanilla CoT still employs LLM to directly answer reasoning questions but provides intermediate reasoning steps before offering the final answer. The first advantage of our method is that **we decompose the challenging task of directly answering reasoning questions into two simpler sub-tasks**: one sub-task is breaking down the original question into simpler sub-questions, and the other sub-task is answering a series of simple questions. **This effectively reduces the complexity of the original task and is more suitable for SLMs**, which is an advantage not present in vanilla CoT. Furthermore, we propose a **step-level PPO**, which is fundamentally different from the token-level PPO in RLHF[1], to further optimize the model's reasoning path. The overall method yields significant gains compared with Vanilla CoT finetuning.
>
> **R2:** The purpose of using PPO is to choose a better reasoning path, but the experimental part only shows the overall effect after using the PPO method. It is recommended to supplement the model accuracy at different stages after dismantling sub-problems for different methods, which can better illustrate the method's effectiveness.
>
> **A:** We select samples that require decomposition into three sub-questions from all test sets and report the accuracy for each sub-question, as shown in the table below. **It is evident that PPO significantly improves the model's performance on reasoning tasks at every step.** Additionally, we found that as the number of steps increases, the performance gain from PPO tends to diminish. We will include the detailed experimental results in the revised version. Thank you for your valuable comments.
> |Method|first step|second step|final step|
> |:-----:|:---:|:----:|:----:|
> |DialCoT-S|60.7|48.8|43.8|
> |DialCoT-S-PPO|63.9|51.2|45.3|
>
> **R3:** Using the dialogue method to reason, the comparison of reasoning speed and time cost compared with the previous method needs to be supplemented.
>
> **A:** In terms of the time cost for reasoning, **the inference speed of our method is comparable to that of SelfAsk[2]**, as both need to generate sub-questions and answers. In terms of the number of iterations, **DialCoT-A requires two iterations, which is comparable to Zero-shot CoT[6]**, while DialCoT-M and DialCoT-S require approximately three and six iterations respectively. Overall, in our methods, DialCoT-A has the fastest inference speed, followed by DialCoT-M, and finally DialCoT-S. The **trade-off between inference time and performance** needs to be considered. If faster inference speed is required, one can opt for DialCoT-A, at the expense of some performance loss. Conversely, if better performance is the priority, DialCoT-S can be chosen, although this would come at the cost of increased inference time. We will include a more detailed report on the inference costs in the revised version. Thank you for your valuable comments.
>
> **R4/5:** Figure 2 is overcrowded, poor readability. Figure 3 is the main method diagram of the paper, which is separated from Figure 2, and it isn't easy to intuitively compare the differences between different methods.
>
> **A:** Thank you for your valuable suggestions. In the revised version, we will place the architecture diagrams of DialCoT-S-PPO and the other methods together to make it easier for readers to compare the differences between them. We will also try our best to reduce the texts in Figure 2 to enhance readability.
>
> **Q1:** If the problem is more complex and the number of rounds of dialogue increases, will there be a significant decrease?
>
> **A:** Firstly, through statistical analysis, we found that on average, a question in our test set needs to be decomposed into 2.7 sub-questions. Secondly, we select samples that require decomposition into three sub-questions from all test sets and report the accuracy for each sub-question, as illustrated in the table below. In this context, CoT-FT refers to the results obtained by replacing the original question with different step-based sub-questions. As shown in the table, as the number of steps increases, i.e., as the complexity of the questions rises, **our method shows a notable performance improvement (~7%) compared to CoT-FT**. On the other hand, although our method's performance does decline as question complexity increases, **the rate of decline is significantly slower compared to vanilla CoT.** We will include the detailed experimental results in the revised version. Thank you for your valuable comments.
> |Method|first step|second step|final step|
> |:-----:|:---:|:----:|:----:|
> |CoT-FT|55.2|41.9|34.7|
> |DialCoT-S|60.7|48.8|43.8|
>
> **Q2:** Some parameters in the RL section are set relatively briefly. For example, why should rm be selected as 0.3 in Table 1 in the reward model? Is it differ from task types?
>
> **A:** We obtained **the best performance on the validation set** when the intermediate reward score was set to 0.3, as determined through **grid search**. Therefore, we chose to set the intermediate reward score at 0.3. **Using the same settings across multiple arithmetic reasoning datasets yielded significant performance gains.** We are in the process of extending our method to common sense reasoning and symbolic reasoning tasks, and we will continue to explore the impact of these hyperparameters on different types of tasks in the future.
>
> **Q3:** Homogenization of task types. As the paper says, "All datasets comprise arithmetic reasoning problems at a primary school level, varying by the entities they incorporate", I doubt the generalization of methods on other tasks.
>
> **A:** In this paper, we **focus on arithmetic reasoning tasks**. We conducted experiments on **four arithmetic reasoning datasets** to validate the effectiveness of our method. These four datasets cover **the most commonly used datasets in arithmetic reasoning-related papers**[3][4][5], sufficiently demonstrating the generalizability of our method for arithmetic reasoning tasks. We plan to further extend our approach to commonsense reasoning and symbolic reasoning tasks in the future.
>
> **Q4:** How is the intermediate step data obtained in the training data, such as how to determine whether the solver answers correctly or incorrectly for a sub-problem decomposed from the decomposer?
>
> **A:** The original GSM8K dataset provides sub-questions and answers for each sample. We directly utilized the training set from GSM8K.
>
> **Q5:** Should the action space of models of different sizes be different because the models have different abilities to decompose problems correctly?
>
> **A:** We believe that the action space should differ for models of varying sizes. When the model is larger, the correct response is more likely to appear among the top few candidates, thereby reducing the space that needs to be explored.
>
> **Q6:** Is the method still effective on more complex datasets that require more reasoning steps and larger action space?
>
> **A:** As can be seen from the table above, **our method remains effective for more complex data that requires additional reasoning steps.** On the other hand, as shown in Figure 6, **as the action space increases, PPO becomes more difficult to converge, consequently affecting the model's performance.** Therefore, it's reasonable to speculate that the performance of our method may decline for problems requiring a larger action space. We plan to continue exploring the performance of our approach on more complex tasks in the future.
>
> [1] Training language models to follow instructions with human feedback, NIPS 2022
>
> [2] Measuring and narrowing the compositionality gap in language models, 2022
>
> [3] Chain-of-Thought Prompting Elicits Reasoning in Large Language Models, NIPS 2022
>
> [4] Self-consistency improves chain of thought reasoning in language models, ICLR 2023
>
> [5] Specializing Smaller Language Models towards Multi-Step Reasoning, ICML 2023
>
> [6] Large Language Models are Zero-Shot Reasoners, NIPS 2022

---

### Official Review · Reviewer_Rgam · 2023-08-05

**Soundness:** 4

**Excitement:**

4: Strong: This paper deepens the understanding of some phenomenon or lowers the barriers to an existing research direction.

**Paper Topic And Main Contributions:**

The authors propose a new approach to elicit more advanced reasoning in smaller LMs, by breaking down the CoT into a dialogue format between a "Decomposer" and "Solver." The authors test whether this format is able to better elicit reasoning in smaller LMs such as Flan-T5.

**Questions For The Authors:**

A. How much of the improvement is due to the proposed prompting method vs fine-tuning of the LM? An ablation study that investigates the proposed method for non-fine-tuned LMs would be interesting, and conversely, one using vanilla CoT with a fine-tuned LM.

 B. Is the improvement in performance due to PPO statistically significant? 2% doesn't seem like a very substantial difference. But the improvement seems to be larger for decoder-only models.

 C. In Figure 4, what is the prompting method used for "FlanT5"?

**Reasons To Accept:**

- The proposed approach is interesting as it leverages the fact that many recent LMs are instruction-tuned on dialogue/chat data, and so it may be a more effective way to elicit reasoning in smaller models.
 - They experiment with Flan-T5, a smaller open LLM, which facilitates reproducibility.
 - The paper is well-written and easy to understand.

**Reasons To Reject:**

- Aside from the fine-tuning and PPO, the approach seems very similar to recent question decomposition prompting approaches, such as SelfAsk. And the improvement due to PPO seems marginal relative to the other aspects of the approach. Would something like SelfAsk + fine-tuning work just as well as the proposed approach?

**Reproducibility:**

5: Could easily reproduce the results.

**Reviewer Confidence:**

4: Quite sure. I tried to check the important points carefully. It's unlikely, though conceivable, that I missed something that should affect my ratings.

**Typos Grammar Style And Presentation Improvements:**

The paper is well-written, with only a few minor errors:

Equation 4: This could be made easier to read by increasing the spacing between some of the variables, and increasing the sizes of the brackets to more appropriately surround the fractions, etc.

Line 400: "We" -> "we"

---

> ### Author Rebuttal · Authors · 2023-08-29
>
> Before reviewing the responses, **please make sure to read the [general response](https://openreview.net/forum?id=VyjNXY2wgi&noteId=sjLPj6ptDw) first.** Thank you very much for your valuable comments. Below are our replies to the questions you have raised and we hope they can address your concerns.
>
> **R1:** Aside from the fine-tuning and PPO, the approach seems very similar to recent question decomposition prompting approaches, such as SelfAsk. And the improvement due to PPO seems marginal relative to the other aspects of the approach. Would something like SelfAsk + fine-tuning work just as well as the proposed approach?
>
> **A:** Firstly, we propose three different forms of DialCoT, among which DialCoT-A is somewhat similar to SelfAsk[1], while **DialCoT-M and DialCoT-S employ entirely different approaches from SelfAsk** for guiding the model to generate intermediate reasoning steps. Secondly, the results of SelfAsk+ fine-tuning can be approximately analogized to the results of DialCoT-A. Compared to DialCoT-A, both DialCoT-M and DialCoT-S achieved a greater performance gain, indicating that **the approach of incrementally generating questions and answers (DialCoT-M/DialCoT-S) is more effective than generating all questions and answers at once (DialCoT-A/SelfAsk).** Finally, our proposed step-level PPO further optimizes the model's reasoning path, which brings more improvement. Based on our experiments shown in Table 2,   the overall method DialCoT-S-PPO is way more effective than DialCoT-A, which indicates SelfAsk+fine-tuning would not work as well as ours. Nevertheless, a more rigorous study should be performed as future research.
>
> **Q1:** How much of the improvement is due to the proposed prompting method vs. fine-tuning of the LM? An ablation study that investigates the proposed method for non-fine-tuned LMs would be interesting, and conversely, one using vanilla CoT with a fine-tuned LM.
>
> **A:** **CoT-FT is the model fine-tuned in the form of vanilla CoT using the same training data as our method.** As seen in Table 2, our method (DialCoT-S) outperforms CoT-FT by **9.1%**, where the improvement can be mainly attributed to the prompting method we propose. On the other hand, since smaller models do not acquire strong reasoning capabilities during the pre-training phase, relying solely on in-context learning without fine-tuning yields poor results in stimulating their reasoning capabilities, as evidenced in [2]. This is also why current research aimed at improving the reasoning capabilities of smaller models[3][4][5] all require fine-tuning. Therefore, we believe that an ablation experiment using few-shot CoT prompting without fine-tuning on smaller models is unnecessary.
>
> **Q2:** Is the improvement in performance due to PPO statistically significant? 2% doesn't seem like a very substantial difference. But the improvement seems to be larger for decoder-only models.
>
> **A:** Our experimental results are the average outcomes across five different seeds, and our test set contains 20,000 questions. A stable increase of 2% means correctly answering an additional 400 questions. Meanwhile, **it's important to emphasize that this improvement is achieved on top of the already substantial performance gains realized by DialCoT-S.** Therefore,  we believe such improvement should NOT be treated as **minor**.
>
> **Q3:** In Figure 4, what is the prompting method used for "FlanT5"?
>
> **A:** In Figure 4, Flan-T5 employs Few-shot CoT[2] prompting. Specifically, the prompt used includes 8 exemplars, and an example of one is as follows:
>
> "Question: Ivan has a bird feeder in his yard that holds two cups of birdseed. Every week, he has to refill the emptied feeder. Each cup of birdseed can feed fourteen birds, but Ivan is constantly chasing away a hungry squirrel that steals half a cup of birdseed from the feeder every week. How many birds does Ivan’s bird feeder feed weekly?
> Answer: The squirrel steals 1/2 cup of birdseed every week, so the birds eat 2 - 1/2 = 1 1/2 cups of birdseed. Each cup feeds 14 birds, so Ivan’s bird feeder feeds 14 * 1 1/2 = 21 birds weekly. So the answer is 21."
>
> We will add detailed information about all prompts we use to the appendix of the revised version. Thanks again for your valuable comments.
>
> [1] Measuring and narrowing the compositionality gap in language models, 2022
>
> [2] Chain-of-Thought Prompting Elicits Reasoning in Large Language Models, NIPS 2022
>
> [3] Large Language Models Are Reasoning Teachers, 2022
>
> [4] Star: Self-taught reasoner bootstrapping reasoning with reasoning, 2022
>
> [5] Specializing Smaller Language Models towards Multi-Step Reasoning, ICML 2023

---

### Meta-Review · Area_Chair_AxhT · 2023-09-18

**Recommendation:** 5

**Metareview:**

This paper proposed methods to decompose CoT with dialog format for smaller models and a step-wise PPO to improve the results further.

**Pros**: The paper is very well written and proposes a very well-motivated method to solve the problem timely with open-source smaller models. (all reviewers)

**Cons**: Some of the figures could be improved (**orRw**)

Overall, the paper presentation is clear, with strong motivations. The proposed method is intuitive, with solid logic behind the design choices. The experiment results are strong compared to baselines using smaller models.

---

### Decision · Program_Chairs · 2023-10-07

**Decision:**

Accept-Main

**Comment:**

This paper proposed methods to decompose CoT with dialog format for smaller models and a step-wise PPO to improve the results further.

**Pros**: The paper is very well written and proposes a very well-motivated method to solve the problem timely with open-source smaller models. (all reviewers)

**Cons**: Some of the figures could be improved (**orRw**)

Overall, the paper presentation is clear, with strong motivations. The proposed method is intuitive, with solid logic behind the design choices. The experiment results are strong compared to baselines using smaller models.